# A meta-ethnographic systematic review of women's experiences of homelessness in high income environments

**Maxine Radcliffe**[1]*, **Anne Cronin**[2], **Diarmuid Stokes**[1], **Matthew J. Douma**[1], **Debra Jackson**[3], **Thilo Kroll**[1], **Kate Frazer**[1]

1 School of Nursing Midwifery and Health Systems, University College Dublin, Dublin, Ireland, 2 School of Medicine, University of Limerick, Limerick, Ireland, 3 School of Nursing Faculty of Medicine and Health University of Sydney, Sydney, Australia

* maxine.radcliffe@ucdconnect.ie

## Abstract

### Background

Homelessness is a significant public policy and health service challenge globally. Often identified as a 'wicked problem' homelessness is hard to define with limited data confirming the exact numbers of people who are homeless due to varying metrics employed many of which likely exclude women by design. Research and policy have primarily focused on the experiences of single men, and the impact of homelessness on women and their experiences of it are not well understood.

### Objective

To synthesise evidence from qualitative studies of homelessness to identify key dimensions of women in high-income countries (HIC) and their experiences navigating lives when homeless.

### Methods

Systematic searches of six databases [MEDLINE, Embase, Global Health, PsycINFO, CINAHL and ASSIA] were completed from 2012 to 8th January 2024. We included peer-reviewed publications published in English reporting primary qualitative data on women's experiences of homelessness in high-income countries only. A review protocol was developed and published. Noblit and Hare's Metaethnography steps guided the synthesis and are reported according to the eMERGe guidelines.

### Findings

Thirty-two studies were identified describing the experiences of 227 women across nine HICs. A conceptual model comprising three themes is presented within a social-ecological theoretical framework within structural and temporal axes of impact

**Data availability statement:** All relevant data are within the manuscript and its Supporting Information files.

**Funding:** The author(s) received no specific funding for this work.

**Competing interests:** The authors have declared that no competing interests exist.

with effects on individual and societal levels. These themes, 1) Precarity, 2) Existing with Risk and Surviving, and 3) Fracturing Identity, describe the implications of homelessness and how the experience of precarity impacts identity and decision-making abilities. The impact of risk arising from violence and exclusion, coupled with descriptions of shame and stigma, presents insight into women's experiences that have hitherto had a limited presence in clinical discourses.

## Conclusions

The evidence in this review highlights the perpetual reporting of a deficit lens on homelessness. Women experiencing homelessness in HICs are a heterogeneous group that is poorly recognised and understood in the literature. It appears that there is a lack of tailored and responsive service availability and that this further perpetuates the structural underpinnings of homelessness, which cluster in highly gendered ways.

---

## Introduction

Homelessness is a significant public policy and health service challenge globally. There is a growing body of evidence that women experience 'multiple exclusion homelessness' (MEH) in ways that are different from their male counterparts [1–3]. 'Multiple Exclusion Homelessness' (MEH) is a term used in practice and policy, particularly in the UK to refer to the many overlapping or interrelated features of extreme exclusion and deprivation and their sequelae, that homeless people often describe as core drivers of their experience of homelessness [1,4].

Much of the existing literature on homelessness relies on definitions that exclude women, since the sampling methodology may require that to be counted, an individual is visibly vulnerable or apparent in specific contexts and at specific times. Therefore, to date, much of the published literature has focused on single younger men who are necessarily more evident in traditional 'homeless settings', for example, individuals bedded down in inner city doorways late at night [5,6]. Whilst women who experience homelessness are not a homogenous group, there has been little focus in the literature on their experiences of homelessness or examination of how this might diverge from these dominant narratives [7]. Women experiencing homelessness have different needs than men and are a significantly underserved cohort across both service delivery approaches and the research and policy spectrum [7–9].

One of the challenges in reviewing what is known on this topic is that homelessness as a concept is difficult to define because 'home' itself is a concept that is highly culturally situated with many possible interpretations. There is no universal agreement around the threshold that constitutes the 'absence' of home and thus homelessness. Political context is also a significant factor in forming local definitions of what homelessness is or is not. The frame of reference of this review is on experiences of homelessness outside of a context where there are known extreme drivers of displacement or material lack, such as war or extreme poverty. In recognition of this,

we only included papers with data from 'high income' contexts as defined by the World Bank [10] in the review. The World Bank established income classification cohorts for analytical purposes in World Development Report [11] and these are widely used as a useful proxy for analysis or basis for comparison of stages of development between countries.

High income countries are taken as a reference point in this review since the socio-economic circumstances of women in these countries can be seen to be broadly comparable. Experiences of homelessness are highly influenced by the prevailing social and economic context in which they occur. The search strategy included in S2 Appendix lists the contemporary World Bank classification of high-income countries [12] at the time of the review.

The common statutory approach across high income countries is to take the view that homelessness is primarily 'rooflessness' which often results in the exclusion of women from sampling and produces data gathered exclusively using witnessed point in time methodologies [6,13].

Underreporting at the statutory level is a significant issue across all contexts and population demographics as it is politically incentivised [14,15]. FEANSTA [16] recently calculated a rough estimate of homelessness in EU as 895,000 people living in ETHOS [European Typology of Homelessness and Housing Exclusion] Light Categories 1–3 (S1 Appendix; [17]). This would equate to a homeless population of 0.174% of the estimated EU wide population of 530 million.

Whilst previous studies have explored women's experiences in relation to homelessness these have been highly located in contextual characteristic terms, this is the first qualitative metasynthesis, to the best of our knowledge, to investigate what is known about women's experiences of homelessness.

The review aimed to develop a rich understanding of what is known about women's experiences of homelessness in high income countries, and a sense of what is not well understood. A study protocol was designed for rigour and published a priori [18].

## Methods

Meta-ethnography is concerned with both examining what is present in research studies, acknowledging from what perspectives these studies are undertaken, and thus how they relate to one another and may be understood synergistically [19]. We chose this review approach as the stages enable an inductive analysis of the literature that both identifies themes already described in the literature and generates new understandings [20]. Constructs or lines of argument developed through the process are representative of the whole dataset but may not be identified in a single study [21].

### Design

The review adhered to Noblit and Hares [19] seven phases of metaethnography. Using this framework allowed us to develop a model of understanding of how women across high income contexts have described their experiences of homelessness in temporal terms and how this relates to both the physical and psychological realities of this journey in the reality of messy complexities that is experience as lived or as told to/witnessed by a researcher.

The framework guiding the review was:

*Population*: Adult women who have experienced or are currently experiencing homelessness in a high-income country [12].

*Intervention*: Exposure to Homelessness. The working definition of Homelessness from the European typology on Homeless and Housing Exclusion ETHOS Light [17].

***Outcomes*: A rich understanding of women's experience(s) of homelessness in high income countries**

**Search methods.** Systematic searches of MEDLINE, Embase, Global Health, PsycINFO, CINAHL and ASSIA were completed for the years 2012–2022. An initial scoping search yielded 5768 records, and a pragmatic decision was therefore made to limit inclusion to a ten year period (2012 to 2022) since this review was required for doctoral research.

The search was further updated 8th January 2024, as the review timeline was extended and publication best practice requires updated searches. The search strategy was refined with support from a UCD library specialist, four main concepts including keywords were searched: "women"; "homelessness"; "high income countries" and "experiences". The same search approach was conducted for all databases, but keywords were coupled with relevant MESH/thesaurus terms where appropriate. The ASSIA search strategy is included as an example in S2 Appendix. The other search strategies are available on request.

**Inclusion and exclusion criteria.** We included peer-reviewed publications published in English that reported primary qualitative data on women's experiences of homelessness in high income countries. We excluded papers that did not include primary qualitative data reported in the published paper. A summary table of those excluded at full text review is included as S6 Appendix.

**Screening.** Title and abstract screening were completed independently by three reviewers (MR, AC, KF). Using Covidence we screened independently 3332 titles and abstracts and completed full text review of 159 papers. The PRISMA diagram in Fig 1 details this process.

**Data extraction.** Data from all papers were extracted in full by MR and data from 50% of papers (n = 17) were independently extracted by two reviewers (AC and MD) into a bespoke spreadsheet to ensure accuracy and consistency of extraction and coding. We read the studies in alphabetical order and MR compiled a bespoke spreadsheet for data extraction including key variables reporting study characteristics and reported qualitative themes. Many of the co-authors are experienced in completing systematic reviews (MD, TK, KF) and guided the development of the data extraction form used in this review.

**Quality appraisal.** The Critical Appraisal Skills Programme [22] (CASP) checklist (S4 Appendix) was used for methodological quality assessment. Following the initial reading, two reviewers (MR and KF) independently conducted the appraisal using the CASP checklist [22]. Each study was assessed for quality and relevance, and discrepancies were discussed. No study was excluded following assessment. S4 Appendix presents the outcome of the CASP assessment process.

**Data synthesis.** Noblit and Hare [19] describe seven stages to a meta-ethnographic synthesis, starting from the initial conceptualisation of the research question through to expressing the findings. We have set out the details used in Table 1.

The metasynthesis process was not linear and involved four of the review authors (MR, KF, AC and TK) reviewing the summary thematic extraction table (S5 Appendix) after reading and rereading the papers. Subsequent meetings in person enabled discussion of construct interrelation and relational interpretations as well as the formation of lines of argument. Initially we organised themes around our own individual impressions of the third order reciprocal constructs. Debate followed and we printed both second order themes (study authors thematic interpretations of their data) and first order excerpts (participant quotes). These were physically printed, arranged and rearranged and then we (MR, KF, TK and AC) discussed and agreed on the third order constructs. This was further revisited in detail through discussions. We tried arranging the third order constructs various ways, for example: temporal journeys through homelessness or a specific focus of the experience of homelessness, e.g., Mothering whilst homeless. We agreed that the experiences described all had multiple individual and societal features that both overlapped and were layered on a spectrum over time. MR then developed a visual model (Fig 3) to convey this understanding.

The methodological heterogeneity of the papers was explicitly discussed as part of the development of each construct during the synthesis process. Greater significance was ascribed by the team to constructs arising from papers where the authors had explicitly described both their positionality and relationality with the research participants and/or those studies that had longer or multiple engagements with women. This was evident in the wide heterogeneity of first order data presented.

**Identification of studies via databases and registers**

**Identification**

Records identified from Databases
(n =3,594* )
 Web Of Science n = 874
 Embase n=256
 Medline n =1,367
 CINAHL n= 56
 PsycINFO n =512
 ASSIA n=529
Registers (n=0)
 Total = 3,594* aggregate of two
 rounds of search in Dec 22 and Dec
 23

Records removed *before screening*:
 Duplicate records removed (n = 272)
 Records marked as ineligible by automation
 tools (n =269)
 Records removed for other reasons (n = 3)

**Screening**

Records screened
(n =3322)

Records excluded**
(n =3164)

Reports sought for retrieval
(n =158)

Reports not retrieved
(n =0)

Reports assessed for eligibility
(n =158)

Reports excluded:124
Reason 1 Wrong outcomes (n = 50)

Reason 2 Wrong study design (n = 6)

Reason 3 Wrong patient population (n = 7)

Reason 4 Data attribution unclear (n=2)

Reason 5 Wrong setting (n=1)

Reason 6 Cannot report women's accounts (n = 5)

Reason 7 Not a peer reviewed publication (n = 15)

Reason 8 Unpublished/not peer reviewed work (n = 3)

Reason 9 Main article not in English language (n = 2)

Reason 10 Pre 2012 (n = 1)

Reason 11 No qualitative data from women
experiencing homelessness (n = 10)

Reason 12 Focus of data presented NOT on
experiences of/relating to homelessness (n= 22)

**Included**

Studies included in review
(n =32)
Reports of included studies
(n = 34)

**Fig 1. PRISMA diagram.**

## Results

The focus of this review was to explore what is known about women's experiences of homelessness. None of the studies included were significantly refutational about other studies in the review. Where the study descriptions of phenomena differed, this was due to contextual specifics. The studies included were very diverse in focus, design, and characteristics. The studies were reciprocal in their description of findings around many of the features of women's experiences of

**Table 1. Stages of the Metaethnography.**

| Stages of Meta-ethnography Noblit and Hare | Process adopted in review |
|---|---|
| **1** *Getting Started* | Conceptualise and refine research question: MR in dialogue with KF and TK. |
| **2** *Deciding what is relevant to initial interest* | Systematic search and screening: see PRISMA. MR undertook two rounds of searching with support from DS and KF, AC, and MD, TK resolved any screening conflicts. |
| **3** *Reading the Studies* | Iterative thorough reading and re reading to describe and understand key concepts in the literature MR, read and re read all the studies many times, KF, AC TK and MD also read all studies for familiarising |
| **4** *Determining how the studies are related* | Identify relationships and abstraction of data: explore and categorise constructs. MR built a bespoke spreadsheet to extract first order (primary qualitative data: women's voices direct quotations) and second order (study authors' conceptual interpretations) themes as well as other study characteristics data. This facilitated recognition similarities and relationships between the thematic data. 17/34 studies double (blind extracted) by MR, KF, AC and MD |
| **5** *Translating the studies into one another* | 'Reciprocal translation': Constant iterative comparison of both the data presented and interpretations of the data between papers, identifying overarching constructs that illustrate and explore the underlying meanings: *loss of agency, the deficit lens.* <br> MR, KF, TK and AC, held a series of meetings to discuss and refine the interpretations. Data were discussed, arranged and rearranged of the first and second order excerpts and themes to create possible third order constructs. |
| **6** Synthesizing translations | Inductive analysis to form a new interpretive context: Third order constructs: e.g., *Identity*: as an overarching theme describing both *individual shame* and *societal stigma*. <br> MR developed a conceptual visual model that was refined following subsequent meetings and discussions with TK, KF. |
| **7** *Expressing the synthesis* | Dissemination of findings: this paper is part of this along process of dissemination with conference presentations |

homelessness described and reported in the studies, for example, experiences of intimate partner violence or reasons for leaving or staying in homelessness.

## Study characteristics

This review synthesises qualitative data from 227 women who participated in 32 individual studies reported in 34 published records from nine high income countries. The geographical distribution of the nine studies by country of origin is displayed in Fig 2.

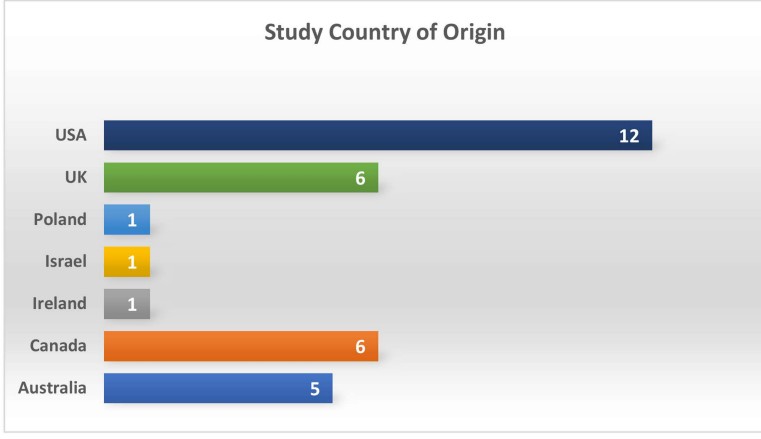

**Fig 2. Study Country of Origin.**

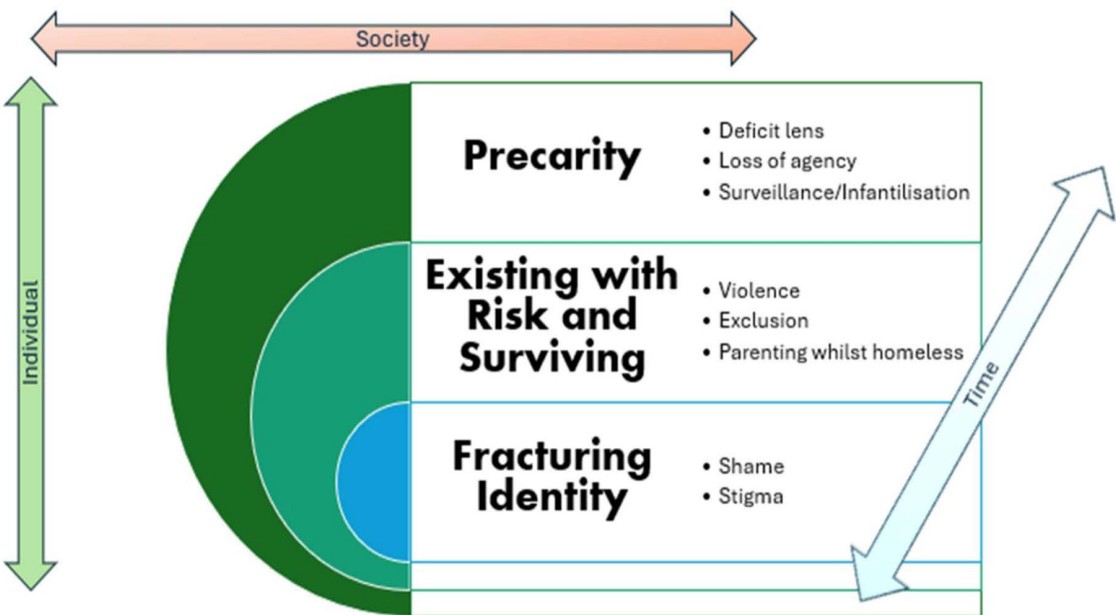

**Fig 3. Women's Experiences of Homelessness.**

A summarised data extraction table is presented in S5 Appendix. Of the 32 studies reported in 34 records in the review, five were mixed methods with additional quantitative data included beyond interviewee demographics. Most studies (28/32) report data from urban locations.

Data collection across the studies in the review report descriptive qualitative methodologies with (n = 19/32) of studies included gathering data through a single 'qualitative' interview with varying approaches to interview technique. Some authors did not describe their data collection methodology beyond referring to 'an interview'. Two studies [23,24] collected data through focus groups and provided detailed descriptions of methodologies used.

A minority of studies (n = 9) [23,25–32] included detailed reflexivity and consideration of the researcher's power and position in developing recruitment strategies and study design. Some studies used ethnographic participant observation and other fieldwork methods as well as interviewing participants [9,25,30,33], and three studies [9,29,33] involved either photovoice/photo elicitation or an arts collaboration component.

Defining homelessness was reported differently across the studies, with (n = 17/ 32) reporting a working definition of the term. The remainder (15/32) did not report a definition of the term. The definitions reported ranged from local statutory definitions to the authors' own working definitions of the concept.

Most studies (n = 28/32) recruited women participants from an urban sampling context, with none recruiting from rural contexts. Four studies report recruitment strategies supporting the recruitment of participants from mixed settings.

## Meta-synthesis outcomes

The inductive process of synthesis in meta-ethnography requires weaving identified threads together into a cohesive narrative or line (s) of argument [19]. We constructed a model to represent our line of argument synthesis with three layers of overarching themes that are constituted by further sub-thematic layers. Fig 3 presents the interrelation and layering of each of these. The reporting of the subsequent themes includes direct reference and quotes [exemplars] from primary studies reported in this systematic review with the identified study number corresponding to this review's reference list.

This interpretation draws on social-ecological theoretical frameworks [34] as the themes that emerged are positioned within structural and temporal axes of impact, with effects on both the individual and societal levels.

*Precarity* is apparent in multifaceted ways throughout the descriptive narratives of the papers in this meta synthesis. There are two major further coherent layers to the narratives in the studies that we have characterised as *Surviving Extreme Risks* and *Identity Fragmentation.* These layers can also be viewed as aspects of precarity from an individual perspective. These concepts form the bedrock of women's narratives about their homelessness experiences from the studies included in this review.

## Precarity: a highly uncertain existence

Precarity is an evolving concept developed through various disciplines of study, notably philosophy, social geography and gender studies [35]. Historically it is rooted in political economy dialectics around lack of labour security and economic domination and exploitation [36,37]. The work of Judith Butler [38] established precarity as a lens through which to focus on understanding the relationship between structural forces and individual and community experiences [35].

It is clear from the participant accounts that experiences of homelessness for women are often explained as powerlessness and subjugation [26,30,39]. Women often reported experiencing homelessness as extreme instability or threat at existential levels. They repeatedly survive significant losses of agency, domination, and exploitation [25,32,40,41].

Precarity, conceptualised thus, provides an intellectual scaffolding to appreciate the effects of intersectionality acting at individual and structural levels [42]. That is to say that women experiences of homelessness are influenced by multiple, often interlocking or intersectional forces that profoundly affect aspects of the experience. For example, an individual may identify as black or white and migrant or local. This, despite policy aspirations, may affect what access they are given to supports and how they might be treated in housing support systems [43].

Recent commentators [44] describe precarity as a critical and emerging social determinant of health. The unflinchingly repetitive and cyclical nature of some of the experiences the women in the review relate, [25,45], and women's lack of control over the trajectory of these is striking [25,46–48]. Women describe their movements and behaviours as being structured both by the institutional settings they were accessing and other individuals, typically hostile men [25,45].

Further, these power relations that result in women lacking agency and facing hostility in wider society are often perpetuated whilst women are experiencing homelessness [25,30,45]. Precarity thus is threaded in multilayered underpinning ways throughout the women's narratives of their experiences of homelessness in all 34 papers in the review.

## Existing with risk and surviving

Women's accounts of their experiences frequently referenced places they were '*allowed to be*' or things they were '*forced to do*' [39,49]. The 'choices' available to women or range of unsatisfactory options/exploitation women describe throughout their narratives involve both individuals and organisations [25,46,47]. Controlling individuals, usually other men, who were threatening or abusive or both are visible and evident from all of women's accounts in the review to the point that the papers themselves at times were harrowing to read and the commonplace nature of this violence almost seemed unremarkable [7,9,30,39]. An example of an experience is presented and is typical of other women's accounts:

> "*He put a knife to my throat, he had tried to strangle me, he had taken all my funds, I was isolated. And he told me that if I ever left, he would hunt me down and shoot me like a dog*" [Hannah in [40]].

Women also described feeling controlled or 'under surveillance' by the available support services such as outreach or shelter workers [7,41,45,47,50]. An exemplar of the exercise of power by 'helping workers' is presented:

 

*"They play this threatening thing. But they hide it. They say, you have to move, otherwise, you have to pay for it. Or if we gave you a temporary accommodation, and you refuse it, we're not going to house you anymore, we're going to close your file, where you going to go with two boys?"* [Jane in [40]].

The price for women of any lack of compliance with demands placed on them is typically highly adverse whether these come from a system or another individual. In the main women describe responding by trying to or conforming to the sanctions imposed or demands leveraged on them with a resulting impact on their own mental health and wellbeing [48,51].

Many women described this complete lack of control over even their own bodies and how this led them to a sense of utter futility and being trapped in a cycle of despair [30,40,41]).

*Cos some of them boys, if they like the look of you, they'll follow you, they won't leave you alone, that's why you have to learn to fight them [...] but then again you'll never beat a man will ya, never. I had one stalking me for two and half years, he smashed our tents down, beat us up [...]he picked me up by me throat, and I went down the police station and told them. Police not bothered though, They're not bothered, at all [...] like anybody could come in them tents, you can't lock them. I even got raped when I was in the tent.* [Tess in [41]].

In terms of individual experiences, women described spending entire days just surviving or engaging in the 'business of homelessness' within a series of endless transiting from each appointment or 'supportive' encounter at different times in diverse locations for no discernible reason [26,46]. Women described the negotiating of time and scheduling that is inherent in shelter living that felt like un unfair arbitrary exercise of power.

*"A client requested to be woken up at four o'clock in the morning, so she could get ready to work at a temporary agency and [the supervisor] Ellen told her that she is not waking anybody up at four o'clock in the morning and if she catches anybody up before six, she's barring them, and then, when the woman tried to ask her why, she turned around and called the police and had her removed… "* [D in [25]].

Participant accounts in many studies also convey the frequently cyclical nature of their experiences of homelessness that can be further entrenched by the demands of the business of homelessness and the women's lack of agency in this [7,26,28,45].

The way women are described or defined in the studies in the review often foregrounds their losses or negative experiences as part of their identity, for example as "abused women" or "homeless women" [39,52].

Framing of women through their deficit(s) is evident in most of the papers in this review; only three studies [9,30,50] report women's abilities or capacities without identifying them with their negative experiences either implicitly or explicitly. Whilst this may be in large part because women are describing highly negative experiences it is important to appreciate that language has a pervasive effect and can reinforce systemic oppression [53,54].

Experiences of violence and abuse are established as a frequent pathway into homelessness for women and children and this was evident throughout participant accounts in the papers reporting on these areas of experience [7,32,39,41,45]. Women in all of the 34 papers describe being on the receiving end of significant episodes of extreme physical violence either whilst experiencing homelessness or prior to it. These studies [7,32,39,41,45] focused more on the manifestations of violence and abuse women experienced including multiple and extended rapes and other seriously injurious behaviours including stalking.

*"My body was his body. He raped me and threw me out, so I couldn't go back. I was badly beaten up; hospitalized for four days. I couldn't see for two days"* [Unnamed woman in Reference [39]].

Women described the risks they experienced as further intensified when trying to maintain their mothering role and responsibilities whilst experiencing homelessness.

> *"I started to come into town into B&Bs, carrying three kids around with me... oh it was horrible, the kids couldn't get to school at all. The kids would have been in these places (hostels) and all. They (hostels) were horrible. We had to be out at 11 o'clock in the morning, walk the streets till 5 or 6 in the evening. It wasn't a nice thing to do with kids. [Viv) in* [11]] *"I was very scared. I went to the place where I've had a storage unit and I knew that place locked their gates at 9. So, I secured the back of the car and put pillows for the kids to sleep. I pretty much stayed up all night watching them (cries). And, I waited until the gate closed. Then I knew we were safe and nobody could come in until 6 the next morning. And we would get up and I would get them cleaned up and ready for school and take them to school. And we'd start over the next day"* [Constance in [46]].

### Fracturing Identity

Women describe feeling forgotten, left out and worthless [23,26,31,41]. Individual level systematic and repeated experiences of exclusion are illustrated in the often-daily mundane processes homeless women experience of being 'othered' intensively with all its contingent consequences. Othering can be briefly described as the development of discourses that uses 'us' and 'them' [55,56]. Historical examples such as colonialism and slavery demonstrate the resultant systemic oppressions that are a direct outcome of othering [55–57]. Women provide vivid descriptions of their experiences of othering and constantly being judged as inferior or 'less than'.

> *'the marginalisation is horrendous'. And, you know if you are homeless, you are homeless for a reason, because you are stupid and it's your fault. Yep. And don't tell people you are homeless"* [Hannah in [58]].

Women describe their self-worth draining away and feelings of valueless and as a 'nonperson' to blame for the circumstances in which they find themselves [7,25,32,45,47,50].

The public disapproval or stigmatisation of homeless people is a relatively unpalatable but commonplace social norm [59]. Further stigma and shame are attached to women who are part of societally excluded groups in even stronger ways [60].

Many of the studies describe how homelessness as an experience, was conceptualised and reacted to at an individual internal level in a way that changed both women's internal narratives about their own identity [28,32] and the external character they were required to present in day-to-day life [30–32,58].

Women created 'survival persona' [30,45] as different identities to cope with the ongoing threats, instability and extreme precarity they were experiencing. This alternate persona they were often forced into, required and promoted coping mechanisms that could be highly self-injurious such as substance abuse, addiction and sex work. Further servicing these coping strategies generated other obligations that maintained instability and homelessness [45,50].

> *"I was couch surfing but there was many a night where I'd have to get out of there because they assume that means sex in bed and rock and roll, you know... Because you owe something. And once you owe something, they can take anything. It's dirty. It's a really ugly, you know the word rape is um, is so misunderstood even as a victim of it because if you're doing it for a place to stay, am I being raped? Or am I f****** him so I can have somewhere to sleep. You know what I mean? Excuse my language. It's a horrendous place to be."* [X in [50]].

Throughout all the studies in the review women described accommodations they tried to undertake to keep themselves or their children 'safe' often enduring abusive experiences for many years [32,39–41,50,61] detail these experiences explicitly with almost all the other study participants referencing this at a minimum in oblique ways.

The sequelae in terms of the psychological consequences of homelessness for women are also clearly articulated and enumerated as experiences of toxic shame and stigma [62] through the negative commentaries or judgements they described experiencing externally [27,33].

Women referenced lengthy and cyclical episodes of homelessness or rooflessness as they became further separated and isolated, for example not reaching out to family networks to avoid feeling shame [32,41,50,61] or ending up entrenched in a pattern or situation which reinforces their exclusion.

## Discussion

In this review, we sought to identify, explore, and synthesize existing qualitative evidence reporting women's experiences of homelessness from studies completed in high-income countries. Exclusion at a structural level appears pervasively evident in both the evidence reported in the review and the policy discourses around homelessness. This may manifest further as deficit thinking or a focus on individual characteristics that may be negatively focused and fail to take account of individual agency or capability in the context of measurement.

The evidence in this review suggests that women experiencing homelessness are likely to be represented minimally in the broader data collected on homelessness in high income countries. This is because many data-gathering approaches undertaken to date rely on frames of reference that are excluding by design. Only three studies in the review did not have recruitment strategies that relied on accessing participants from established homeless services with access criteria. This likely means that the women included in the studies had selected specific characteristics, excluding those not accessing services. Only 14 of the 32 studies in the review had an agreed sample reference definition for homelessness. Whilst this is somewhat inevitable in looking for a sample from a population that is *hard to reach*, it is noteworthy in terms of drawing conclusions about women's experiences based on this evidence base.

Homelessness is evidenced in all the papers in the review as a powerful and multimodal experience of exclusion and difference or 'othering' which is established as having particularly accentuated stigmatising effects for women [13,52,54,56]. These experiences intersect at both an individual and societal level with the internal and external dialogues of exclusion and disadvantage in a similar way to other group or individual experiences of difference or othering such as coming out. Intersectionality [42,63,64] therefore holds significant conceptual importance in examining women's perspectives on their experiences both as way of thinking about the multiple dimensions that influence homeless women's identities as they themselves construct them and in understanding how this plays out in the attendant oppressions they experience or privileges they are accorded.

The review findings suggest that homelessness may entrench experiences of exclusion and disparity beyond the material realities women experience of not having a safe place to call home. For women the legacies of this may be manifold, including highly adverse psychological sequelae such as toxic shame and trauma, very significant risk to life and limb, and loss of family and other significant relationships.

### Strengths, limitations and reflexivity

Women experiencing homelessness are a hard-to-reach group in terms of research as they are often hidden from services [7,45] and sampling strategies, especially those that are point in time, or involve seeking participants from an already selected group who are often, for example, accessing a particular support centre. This may introduce a layer of power relations that may not be immediately apparent but could significantly affect the research outcomes.

The papers included in the review presented data gathered from women in predominantly highly urban contexts, thus rural homelessness women's experiences are not represented. This limitation should be noted when drawing any wider conclusions based on the findings about women experiencing homelessness.

We chose to utilise the ETHOS typology (Appendix 1,19) as a base reference definition for homelessness in our search strategy. Most of the women in the studies included (29/32) were recruited via services with specific criteria for

access that excludes large sections of the wider population of women who would fit under that typology even taken at the most reductive level. Secondly, recognition of rough sleeper status in either London (UK) or Dublin (Ireland) requires a visible bedded down presence on the street that is witnessed by a specific officer with the powers to 'verify' homelessness [16] and this may be different for the US or other HIC studies included in this review. Maintaining the type of visible presence required to obtain this recognition is costly in terms of risk for women and thus many choose to eschew this.

The significant definitional variability in how the term homelessness was used within the papers reviewed and the absence in some papers of a definition is somewhat to be expected. The papers included were from a broad range of disciplinary origins, including for example sociology or ethnography where a focus on setting out a boundaried or *limiting* definition of homelessness would be counter to the entire epistemological approach of the research. Homelessness itself is a highly situated or *culturally constructed* concept rather than a binary absence or presence. The focus of the review was on what women themselves had to say about homelessness and their experiences of this and thus whilst it is noteworthy, particularly from a policy perspective that there are challenges defining homelessness, we have chosen to include studies without a published included definition since they clearly presented rich qualitative data on women identifying as homeless.

We excluded papers not published in the English language and we acknowledge this impact on the evidence base. However, the evidence in this review, synthesised using robust frameworks, is novel in methods and highlights critical gaps in evidence for future research.

## Conclusion

Women experiencing homelessness in high income contexts reported in this systematic review of evidence are a heterogeneous group that is poorly recognised and understood in the literature. It appears that there is a lack of tailored and responsive service availability and that this further perpetuates the structural underpinnings of homelessness, which cluster in highly gendered ways.

Women experiencing homelessness face multiple overlapping disadvantages, and the character of experiences that lead to their homelessness and maintain them in this precarious and often cyclical state are highly insidious. Within this body of evidence there are some consistent aspects to the experience as described in our socioecological model of a highly precarious existence overlaid with significant levels of personal risk and highly adverse physical and psychological sequelae. This is coupled with individual experiences of extreme prejudice and isolation, which have a profound impact on individual ability and capability to function in both daily life and along broader societal norms.

We have endeavoured to synthesize a large volume of qualitative data from disparate studies across diverse contexts and adopting varied research methodologies. Despite the breadth and detail of this canvas, the women's voices in the 32 studies included in this review are very clear in describing the narrative/telling a story of systematic exclusion and structural violence that has had a profound effect on their individual daily lives in the short and long term.

Women experiencing homelessness have different needs to men and are a significantly underserved cohort across both service delivery approaches and the research and policy spectrum. Research that incorporates both acknowledgement and appreciation of the role of power differentials inherent in the process of research, as well as the focus of study, is critical to moving the evidence base further in ways that are likely to contribute fruitfully to improving policy and practice responses to homelessness. As researchers and practitioners, we must examine our approaches as re(producers) of discourse about women to avoid entrenching deficit-driven perspectives.

## Supporting information

**S1 Appendix. ETHOS Light Typology of Homelessness [15].**
(PDF)

**S2 Appendix. ASSIA Search Strategy.**
(DOCX)

**S3 Appendix. Study Characteristics Table.**
(DOCX)

**S4 Appendix. Completed CASP Checklist.**
(DOCX)

**S5 Appendix. Summary thematic extraction table.**
(DOCX)

**S6 Appendix. Table of Studies Excluded at Full Text Review.**
(DOCX)

**S7 Checklist. PRISMA Checklist.**
(DOCX)

## Author contributions

**Conceptualization:** Maxine Radcliffe, Thilo Kroll, Kate Frazer.

**Data curation:** Maxine Radcliffe, Anne Cronin, Matthew J. Douma, Thilo Kroll, Kate Frazer.

**Formal analysis:** Maxine Radcliffe, Anne Cronin, Thilo Kroll, Kate Frazer.

**Methodology:** Maxine Radcliffe, Diarmuid Stokes, Thilo Kroll, Kate Frazer.

**Supervision:** Thilo Kroll, Kate Frazer.

**Writing – original draft:** Maxine Radcliffe.

**Writing – review & editing:** Maxine Radcliffe, Debra Jackson, Thilo Kroll, Kate Frazer.

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
