## [Decision Letter · Decision Letter 0]

12 Feb 2025

Dear Dr. Radcliffe,

Thank you for submitting your manuscript to PLOS ONE. After careful consideration, we feel that it has merit but does not fully meet PLOS ONE’s publication criteria as it currently stands. Therefore, we invite you to submit a revised version of the manuscript that addresses the points raised during the review process.

We look forward to receiving your revised manuscript.

Kind regards,

Andrea Cioffi

Academic Editor

PLOS ONE

Journal Requirements:

2. We noted in your submission details that a portion of your manuscript may have been presented or published elsewhere. [An abstract representing early findings that I am presenting to the UK Public Health conference in November 2024 will be published in the book of conference abstracts in the Lancet. This is an abstract only

Please clarify whether this conference proceeding was peer-reviewed and formally published. If this work was previously peer-reviewed and published, in the cover letter please provide the reason that this work does not constitute dual publication and should be included in the current manuscript.

3. We note that there is identifying data in the Supporting Information file < Appendix3Character.docx, Appendix_4_CASP.docx and Appendix 5-SummaryThematicReview.docx>. Due to the inclusion of these potentially identifying data, we have removed this file from your file inventory. Prior to sharing human research participant data, authors should consult with an ethics committee to ensure data are shared in accordance with participant consent and all applicable local laws.

-Location data

Additional guidance on preparing raw data for publication can be found in our Data Policy (https://journals.plos.org/plosone/s/data-availability#loc-human-research-participant-data-and-other-sensitive-data ) and in the following article: http://www.bmj.com/content/340/bmj.c181.long .

Please remove or anonymize all personal information (Name, Date), ensure that the data shared are in accordance with participant consent, and re-upload a fully anonymized data set. Please note that spreadsheet columns with personal information must be removed and not hidden as all hidden columns will appear in the published file.

4. As required by our policy on Data Availability, please ensure your manuscript or supplementary information includes the following:

Reviewers' comments:

Reviewer's Responses to Questions

**Comments to the Author**

1. Is the manuscript technically sound, and do the data support the conclusions?

Reviewer #1: Yes

Reviewer #2: Yes

2. Has the statistical analysis been performed appropriately and rigorously?

Reviewer #1: Yes

Reviewer #2: N/A

3. Have the authors made all data underlying the findings in their manuscript fully available?

Reviewer #1: Yes

Reviewer #2: Yes

4. Is the manuscript presented in an intelligible fashion and written in standard English?

Reviewer #1: Yes

Reviewer #2: Yes

Reviewer #1: The paper fills in a gap in our understanding of homelessness among women and the experiences they go through. It is a valuable resource in that sense.

A few points merit noting :

- There are different metrics of homelessness in the nine HICs [ lines 11,189-192]. As pointed out at lines 369-370, only 14 of 32 studies had had agreed sample reference for homelessness. This could skew our understanding somewhat, but as given in the London/Dublin example at the end, this is driven by the laws unique to each country. This has been circumvented to some extent in the paper and a standardized definition has been used[ line 70 and others], however, this point does remain as an issue to be addressed

- The studies had a total of 227 women in 9 HICs. Of these, one third are in USA [line 174]. It is possible that the interpretation may be skewed by what happens in US, which needs to be addressed upfront. It is possible that a woman in Dublin may have a different experience relating to shelters- for example, which may not get captured, being the experience of a small minority of the sample. Besides, it is also possible that 227 data points may not be a large enough sample to derive a common conclusion for all the nine HICs.

- The paper mentions the power dynamics between researchers and women. In all the studies, the expectations from the persons asking the questions may be different. In fact the designation of the researcher, whether NGO, shelter staff, volunteer etc. may elicit responses unconsciously tailored to their expectations, since the women are already in a vulnerable position. These unsaid expectations could thus very well colour the responses, as is well documented in case of qualitative studies. It is not clear if indeed in some studies the power dynamics could have played out in the conclusions derived. This factor, indicating the kind of persons/researchers who did the studies, may be useful as a note for the readers.

Reviewer #2: Reviewer Comments

Thank you for this important research that aims to shed light on the invisibility of women’s homelessness in high-income countries. Because the majority of women deal with issues of abuse, shame, and stigma regarding their household dynamics, they are unlikely to be seen at public shelters or drop-ins, resulting in a lot of undocumented cases of women’s homelessness issues. The few that end up in shelters or seek support face structural challenges and may be excluded from research about homelessness. This meta-ethnography analysis is timely and critical in adding to the literature about women and homelessness in high-income countries. While the study has merit and contributes to knowledge, authors must address the following comments to get the manuscript to publication.

**Introduction**

The authors need to expand the introduction by providing a justification for their chosen context—high-income countries—and addressing why we need to understand the gendered homelessness situation in these countries. What is currently known about this issue in the study contexts, and why is it important?

Line 45 – please expand ‘multiple exclusion homelessness (MEH) and provide examples.

Lines 64-68 – why high-income countries; explain. While it is essential that the authors mention using the World Bank classification of high-income countries, they need to state the countries under review.

**Methods**

The authors claim to have used Noblit and Hare's (1998) meta-ethnography framework but fail to explain how they followed the seven steps Noblit and Hare provided. The authors need to expand the methods section to include what they did at every stage. If the authors did otherwise, they must justify and explain any modification.

Lines 108-111: The authors said they completed searches for 10 years (2012-2022) and then decided to expand the search to include literature from 2022 through January 2024 but did not explain why this was done. First, why did you decide on 10 years? Second, what led to the inclusion of additional data? These things need to be clearly explained.

Lines 119-120 – were there specific things the author excluded and/or included, and did any decision about the inclusion/exclusion criteria affect the data analysis?

**Results**

The authors decided to capture some information about the methods under the results section, which affected the flow of the paper. I suggest moving information from lines 122-158 to the methods section and making a link to Noblit and Hare’s framework.

Line 126 – please explain this statement: ‘34 papers reporting on 32 studies in the review.’ Does this statement suggest two papers written in the same context? If so, clearly state that. The reverse is captured in Figure 1.

Line 159 – is where the result section should begin.

Lines 124-126—Please review the numbers and compare them to the ones in the figure. There are discrepancies. For example, the authors said they screened 3332, but in the diagram, 3594 is captured. Also, they said the final list under review is 159, but in the figure, 158 is captured. Please make the necessary corrections. But if this is not an error, the authors should explain this discrepancy.

Line 171 – ‘five were mixed methods’ – but Figure 1 records 2. Which one is correct?

Line 169 – list the 9 high-income countries.

Lines 202-203 – the title of the figure/image should come under the image.

Lines 205-207 – there should be a signal to the theoretical framework in the Introduction section. This should tie in with the lit review on the motivation for the study; please revise accordingly.

Line 223 – there is something missing in the sentence…’they survive…’

Lines 247-249 – there should be a note about the use of quotes either in the methods section or the first paragraph in the results section.

**Discussion**

How has the social ecological model helped explain the results; that is missing in the discussion section.

**Do you want your identity to be public for this peer review?** For information about this choice, including consent withdrawal, please see our Privacy Policy

Reviewer #1: No

Reviewer #2: No

---

## [Author Response · Author response to Decision Letter 1]

20 May 2025

1. Thank you for your comments. We have checked the paper thoroughly and the have adhered to the style and submission guidelines

2. Thank you for your query. We presented a peer reviewed abstract at a UK Public Health Conference. The abstract was published as part of the conferencing preceedings. It is not the same abstract submitted with this paper. The abstract submitted to the conference was preliminary findings and since there was an updated search. In total the abstract was 300 words and is not in any way a publication of the details and findings in this full text paper.

3. Thank you for your query. We presented a peer reviewed absrtact at a UK Public Health Conference. The abstract was published as part of the conferencing proceedings. It is not the same abstract submitted with this paper. The abstract submitted to the conference was preliminary findings and since there was an updated search. In total the abstract was 300 words and is not in any way a publication of the details and findings in this full text paper.

4. Thank you for your query. We are submitting the CASP quality assessment and the Table of Characteristics again. We have labelled these files accordingly. We are surprised the data files were excluded as none of them contain any new data. This is a systematic review and as such reports on previously published papers. Part of the processes in completing a systematic review are to present a Table of Characteristics and a Table of quality assessment. The table of data extraction in this review are specifically for qualitative data already published. A meta ethnography re analyses the published analyses in primary published papers. There seems to be some misunderstanding of what a systematic meta ethnography review involves. We confirm there is no new data, this is not a research study but is a systematic review and as such we re-analyse the outcome data reported in the papers that have been assessed for inclusion. We noted that PLOS One has published a meta ethnography from one of our team and that is why we selected the journal for submission. https://journals.plos.org/plosone/article?id=10.1371/journal.pone.0257194 We disagree with the comments and reviewer requirements as they are not referring to data extraction processes used in a systematic review e.g. ID number is the first author of the included paper and year of publication for example. We have amended the title of the paper to include the words systematic review to enable clarification. One of our team is a Cochrane trained author and the standards of reviewing in this review are of this standard. We disagree that our review is unethical or contra to research integrity. The data in this review are already available in the published papers that are included. Please can this misunderstanding be corrected with the reviewer. We have now included the name and reference number in brackets following the inclusion of an exemplar quotations within the results section of this systematic review. The quote is as per published in the the primary author as per the reference number.

5. Thank you and we have provided a response as noted above.

6. Thank you for your comment and we apologise for our oversight. We have added text in the introduction to clarify why HIC were included. We have also strengthened the text in our inclusion criteria to clarify as well. We have expanded the text to explain MEH on lines 43 46 - page 2 of paper.

7. Thank you for your comment and we have added additional text on page 3 of the paper LN 65 to 72 to expand the information for readers. We appreciate the inclusion of our reference only previously.

8. Thank you for your comments. The rationale for 10 years was pragmatic and the updated search was because it is best practice to submit a paper with as recent as search - within 12 months.

9. Thank you for your comment and we have amended this section LN 122-124 and added reflections in ln 406-408

10. Thank you for noting our error and we have included the data in the methods as suggested. We will also add the step of the framework to our meta synthesis section in the results. See LN 147-166

11. Yes that is correct. There are 32 unique studies and two with multiple reporting papers.

12. Thank you for your advice and we agree with your suggestion this has been amended and now starts on Ln167

13. Thank you for noting and we have checked the PRISMA flow diagram and all totals. We apologies as an older erroneous version had been included. We have provided a correct PRISMA flowchart using the correct template from Equator Network resources.

14. Thank you and apologies for our error in omitting this text, these are now displayed in Fig 2.

15. Thank you and we have revised the text and PRISMA flow chart for consistent reporting.

16. Thank you and we have revised the position of the title

17. Thank you for noting and we have revised the text.

18. Thank you and we have revised the methods to note that quotes from published and included studies in this review are included. We have clearly included quotes in the text and the referenced the original source papers.

19. Thank you for your comments. We have added some reflections on this this

20. Thank you for your comment and we agree. We have added this text into the discussion section.

21. Thank you for noting our error and we have included the steps and details in the methods section within a detailed table to explain the processes used transparently.

22. Thank you for your suggestions and we have revised the text as noted and updated in this section.

23. Thank you for your comments and we agree that acknowledging power differentials inherent in research is important for readers particularly as it impacts research conduct. We have strengthened our statements and please see page 12 LN 375-380

24. Thank you for highlighting and we have incorporated suggestions into our discussions and limitations sections

Editor and Reviewer Comments

Author Comments

1. Your abstract cannot contain citations. Please remove any existing citations from the abstract section. You may only include citations in the body text of the manuscript, and please ensure that they remain in ascending numerical order on first mention.

Thank you for your suggestions and we have revised the text as noted and updated in this section.

2. Please ensure that you refer to Figure 2 in your text as, if accepted, production will need this reference to link the reader to the figure.

Thank you for your suggestions and we have revised the text as noted and updated in this section.

As required by our policy on Data Availability, please ensure your manuscript or supplementary information includes the following:

1. A numbered table of all 3322 studies identified in the literature search, including those that were excluded from the analyses.

Many thanks for your suggested data availability comment. We understand the requirement for all empirical data reporting. However, this is highly unusual request and is not required for Cochrane systematic reviews per their MECIR documents and PRISMA reporting. The normal requirements for any review Cochrane or not, as per PRISMA guidelines is to provide a PRISMA table with the reasons for excluding full text papers. We have completed this information.

We have provided a supplemental file with the list of 124 citations and the linked reason for exclusion as captured in the Covidence platform. This is what is captured in this search platform . We would request Editorial advice to understand why an impossible ask is placed on ourselves as we would have to revisit the entire 3322 papers that we undertook title and abstract review only of to provide this.. We wish to publish in PLOS One and are grateful for considered response.

---

## [Decision Letter · Decision Letter 1]

14 Aug 2025

Dear Dr. Radcliffe,

**Please address also these issues:**

All included studies come from urban or predominantly urban contexts; rural homelessness experiences are absent. Please, explicitly acknowledge this limitation in the Discussion, outlining implications for transferability of findings.

Almost half the included studies do not define “homelessness,” potentially undermining synthesis consistency.

Hence, provide a critical reflection on how definitional variability may affect comparability, perhaps including a sensitivity analysis excluding studies without definitions.

There is a large variation in data collection methods (single interview, ethnography, photovoice) and absence of reflexivity in most included studies. Please, discuss explicitly how methodological heterogeneity was handled in the synthesis to mitigate bias.

The time restriction (10 years) is described as “pragmatic” but lacks a strong methodological justification beyond contemporaneity. Please, provide stronger rationale or explore inclusion of older, high-quality studies in a sensitivity check.

You state there were no significantly refutational studies, but this is unexpected in qualitative synthesis.

So, revisit dataset to ensure potentially divergent findings are not being subsumed into dominant narratives; even subtle contradictions can enrich the synthesis.

Intersectionality is acknowledged but underdeveloped; ethnicity, migration status, and disability receive little explicit attention despite likely importance. Please, incorporate more targeted analysis of how overlapping identities shape the homelessness experience.

We look forward to receiving your revised manuscript.

Kind regards,

Andrea Cioffi

Academic Editor

PLOS ONE

Journal Requirements:

Reviewers' comments:

Reviewer's Responses to Questions

**Comments to the Author**

Reviewer #1: All comments have been addressed

Reviewer #3: All comments have been addressed

Reviewer #4: All comments have been addressed

2. Is the manuscript technically sound, and do the data support the conclusions?

Reviewer #1: Yes

Reviewer #3: Yes

Reviewer #4: Yes

3. Has the statistical analysis been performed appropriately and rigorously?

Reviewer #1: N/A

Reviewer #3: Yes

Reviewer #4: I Don't Know

4. Have the authors made all data underlying the findings in their manuscript fully available?

Reviewer #1: Yes

Reviewer #3: Yes

Reviewer #4: Yes

5. Is the manuscript presented in an intelligible fashion and written in standard English?

Reviewer #1: Yes

Reviewer #3: Yes

Reviewer #4: Yes

Reviewer #1: The revised paper is more comprehensive and addresses the concerns raised earlier. it can be published in it's present form as it addresses a key area of concern

Reviewer #3: The authors succeeded in selecting study tiopic and up to my observtions, they followed PLOS ONE protocols. In revieing the revised manuscript, the authors responded well to comments and revised accordingly.

Reviewer #4: The author has incorporated all suggestions and comments. The paper could be accepted for publications.

**Do you want your identity to be public for this peer review?** For information about this choice, including consent withdrawal, please see our Privacy Policy

Reviewer #1: No

Reviewer #3: No

Reviewer #4: **Yes:** Dr. JAYANTA KUMAR BASU

---

## [Author Response · Author response to Decision Letter 2]

8 Sep 2025

Dear Editor,

Many thanks for the welcomed decision and the positive responses from all three peer reviewers. We appreciate the opportunity to strengthen the methodological reporting and enhancing the limitations to our review.

Please see the responses to the points raised by the reviewers as follows

1) All included studies come from urban or predominantly urban contexts; rural homelessness experiences are absent. Please, explicitly acknowledge this limitation in the Discussion, outlining implications for transferability of findings.

Response. Thank you we have added this on page 13.

2) Almost half the included studies do not define “homelessness,” potentially undermining synthesis consistency.

Hence, provide a critical reflection on how definitional variability may affect comparability, perhaps including a sensitivity analysis excluding studies without definitions. Thank you and we have added a critical reflection on this matter on page 14.

3) There is a large variation in data collection methods (single interview, ethnography, photovoice) and absence of reflexivity in most included studies. Please, discuss explicitly how methodological heterogeneity was handled in the synthesis to mitigate bias. Thank you we have added this to the discussion on page 7.

4) The time restriction (10 years) is described as “pragmatic” but lacks a strong methodological justification beyond contemporaneity. Please, provide stronger rationale or explore inclusion of older, high-quality studies in a sensitivity check. Response. Thank you for this perspective. Our review over a 10 and now longer during peer review is a substantial body of evidence of women’s experiences. We did not aim to report studies from inception and similarly to other meta synthesis reported in PLOS One journal a limitation on years of search exists as we have clearly reported in the limitations section. We reject the view that another search is required. We have added additional text that briefly explains that our initial search yielded over 5768 records and the rational for the decision at that point to restrict searches to 2012 onwards. Our current search includes 12 years of data. This review was unfunded and is part of a doctoral research study. We identified examples of similar systemic reviews in PLOS One specifically that have restricted years and present a few both demonstrating meta-analysis and meta synthesis.

2024: limited search and no limitations section reported. https://journals.plos.org/plosntds/article?id=10.1371/journal.pntd.0012718

2025: search years limited to 5 years. https://journals.plos.org/globalpublichealth/article?id=10.1371/journal.pgph.0004483

2025: 10 years search https://journals.plos.org/globalpublichealth/article?id=10.1371/journal.pgph.0004272

2025: 15 year search 2009 to 2024 https://journals.plos.org/plosone/article?id=10.1371/journal.pone.0322796

2025: 10 years and no limitations section.

https://journals.plos.org/plosone/article?id=10.1371/journal.pone.0325683

5) You state there were no significantly refutational studies, but this is unexpected in qualitative synthesis. So, revisit dataset to ensure potentially divergent findings are not being subsumed into dominant narratives; even subtle contradictions can enrich the synthesis.

Intersectionality is acknowledged but underdeveloped; ethnicity, migration status, and disability receive little explicit attention despite likely importance.

Response. Thank you we have revisited our analysis and our interpretation is correct. We have highlighted the paucity of detailed qualitative reporting from the collected studies including women who experience homelessness. We are transparent in our reporting and we have added to the discussion regarding intersectionality on page 9 and 20.

6) Please, incorporate more targeted analysis of how overlapping identities shape the homelessness experience. Response. Thank you and we have further added further commentary on Page 9 .

7) Certain passages are emotionally powerful but verge on advocacy rather than analysis. Retain authenticity of participant voice but balance with analytical neutrality to align with journal expectations. Response. Thank you for highlighting and we have revised the tone of some passages to address this .

---

## [Decision Letter · Decision Letter 2]

7 Dec 2025

A Meta-ethnographic systematic review of women's experiences of homelessness in high income environments.

PONE-D-24-50809R2

Dear Dr. Radcliffe,

We’re pleased to inform you that your manuscript has been judged scientifically suitable for publication and will be formally accepted for publication once it meets all outstanding technical requirements.

Kind regards,

Andrea Cioffi

Academic Editor

PLOS One

Additional Editor Comments (optional):

Reviewers' comments:

Reviewer's Responses to Questions

**Comments to the Author**

Reviewer #1: All comments have been addressed

Reviewer #4: All comments have been addressed

2. Is the manuscript technically sound, and do the data support the conclusions?

Reviewer #1: Yes

Reviewer #4: Yes

3. Has the statistical analysis been performed appropriately and rigorously?

Reviewer #1: Yes

Reviewer #4: Yes

4. Have the authors made all data underlying the findings in their manuscript fully available?

Reviewer #1: Yes

Reviewer #4: Yes

5. Is the manuscript presented in an intelligible fashion and written in standard English?

Reviewer #1: Yes

Reviewer #4: Yes

Reviewer #1: i find that the changes suggested earlier have been accepted and addressed. the current article is more comprehensive and presents the case better than the earlier version. the qualitataive data and womens experience add richness and context.

Reviewer #4: Most of the comments and recommendations are incorporated in the revised manuscript. The paper could be considered for publications.

**Do you want your identity to be public for this peer review?** For information about this choice, including consent withdrawal, please see our Privacy Policy

Reviewer #1: No

Reviewer #4: **Yes:** Dr. JAYANTA KUMAR BASU

---

## [Editor Report · Acceptance letter]

PONE-D-24-50809R2

PLOS One

Dear Dr. Radcliffe,

I'm pleased to inform you that your manuscript has been deemed suitable for publication in PLOS One. Congratulations! Your manuscript is now being handed over to our production team.

Kind regards,

on behalf of

Dr. PLOS Manuscript Reassignment

Staff Editor

PLOS One